

# Skeletal muscle mitochondrial volume and myozenin-1 protein differences exist between high versus low anabolic responders to resistance training

Michael D. Roberts[1], Matthew A. Romero[1], Christopher B. Mobley[1], Petey W. Mumford[1], Paul A. Roberson[1], Cody T. Haun[1], Christopher G. Vann[1], Shelby C. Osburn[1], Hudson H. Holmes[1], Rory A. Greer[1], Christopher M. Lockwood[2], Hailey A. Parry[1] and Andreas N. Kavazis[1]

[1] School of Kinesiology, Auburn University, Auburn, AL, USA
[2] Lockwood, LLC, Draper, UT, USA

Corresponding author
Michael D. Roberts,
mdr0024@auburn.edu

## ABSTRACT

**Background:** We sought to examine how 12 weeks of resistance exercise training (RET) affected skeletal muscle myofibrillar and sarcoplasmic protein levels along with markers of mitochondrial physiology in high versus low anabolic responders.

**Methods:** Untrained college-aged males were classified as anabolic responders in the top 25th percentile (high-response cluster (HI); $n = 13$, dual x-ray absorptiometry total body muscle mass change ($\Delta$) = +3.1 ± 0.3 kg, $\Delta$ vastus lateralis (VL) thickness = +0.59 ± 0.05 cm, $\Delta$ muscle fiber cross sectional area = +1,426 ± 253 $\mu m^2$) and bottom 25th percentile (low-response cluster (LO); $n = 12$, +1.1 ± 0.2 kg, +0.24 ± 0.07 cm, +5 ± 209 $\mu m^2$; $p < 0.001$ for all $\Delta$ scores compared to HI). VL muscle prior to (PRE) and following RET (POST) was assayed for myofibrillar and sarcoplasmic protein concentrations, myosin and actin protein content, and markers of mitochondrial volume. Proteins related to myofibril formation, as well as whole lysate PGC1-α protein levels were assessed.

**Results:** Main effects of cluster (HI > LO, $p = 0.018$, Cohen's $d = 0.737$) and time (PRE > POST, $p = 0.037$, Cohen's $d = -0.589$) were observed for citrate synthase activity, although no significant interaction existed (LO PRE = 1.35 ± 0.07 mM/min/mg protein, LO POST = 1.12 ± 0.06, HI PRE = 1.53 ± 0.11, HI POST = 1.39 ± 0.10). POST myofibrillar myozenin-1 protein levels were up-regulated in the LO cluster (LO PRE = 0.96 ± 0.13 relative expression units, LO POST = 1.25 ± 0.16, HI PRE = 1.00 ± 0.11, HI POST = 0.85 ± 0.12; within-group LO increase $p = 0.025$, Cohen's $d = 0.691$). No interactions or main effects existed for other assayed markers.

**Discussion:** Our data suggest myofibrillar or sarcoplasmic protein concentrations do not differ between HI versus LO anabolic responders prior to or following a 12-week RET program. Greater mitochondrial volume in HI responders may have facilitated greater anabolism, and myofibril myozenin-1 protein levels may represent a biomarker that differentiates anabolic responses to RET. However, mechanistic research validating these hypotheses is needed.

# INTRODUCTION

Numerous studies have reported resistance exercise increases both whole/mixed fraction muscle protein synthesis (MPS) and myofibrillar protein synthesis (MyoPS) rates several days following a single exercise bout (*Damas et al., 2016*; *Mitchell et al., 2012*; *Phillips et al., 1997*, *1999*; *Wilkinson et al., 2008*). Several studies have also reported weeks to months of resistance exercise training (RET) increases muscle fiber cross sectional area (fCSA) (*Mitchell et al., 2013*; *Mobley et al., 2017*; *Petrella et al., 2008*; *Reidy et al., 2016*; *Staron et al., 1994*). These parallel findings have led to a general consensus that RET-induced increases in fCSA likely coincide with increased myofibrillar protein content. The addition of myofibrils to pre-existing myofibrillar structures involves coordinated actions from proteins such as alpha-actinin 2 (ACTN2), myozenin 1 (MYOZ1), myotilin (MYOT), and Sorbin and SH3 Domain Containing 2 (SORBS2) (*Sanger et al., 2002*). This process has been observed in rapidly growing cardiomyocytes (*LoRusso et al., 1997*), skeletal muscle myotubes (*White et al., 2014*), developing zebrafish (*Sanger et al., 2009*), and embryonic chicken heart rudiments (*Ehler et al., 1999*). Given that resistance exercise acutely upregulates MyoPS, it seems logical RET would upregulate these genes in order to increase myofibril protein content in hypertrophying muscle fibers.

From a bioenergetics perspective RET-induced increases in MPS and MyoPS are energetically costly given upwards of four ATP molecules are required per peptide bond synthesized (*Stouthamer, 1973*). Thus, increases in mitochondrial function or volume are likely needed to sustain muscle growth during RET due to the increased energy demand required for intracellular protein accretion. Contrary to this hypothesis, a recent review by *Groennebaek & Vissing (2017)*, which included 16 studies examining how chronic "high load" RET affected markers of mitochondrial volume in whole-tissue lysates, cited 14 of these studies observed no change or a decrease in these biomarkers. While this report suggests RET likely does not increase markers of mitochondrial volume, it remains possible that high anabolic responders to RET may experience greater increases in biomarkers related to mitochondrial volume in order to better support anabolism.

We recently published a study examining skeletal muscle biomarkers related to ribosome biogenesis, inflammation, and androgen signaling that were differentially expressed between high versus modest and low anabolic responders following a 12-week full body RET program (*Mobley et al., 2018*); notably, vastus lateralis (VL) thickness changes was the clustering variable. Herein, we adopted a refined approach similar to *Davidsen et al. (2011)* in order to define high versus low anabolic responders in these subjects based upon three hypertrophic indices including total muscle fCSA, VL thickness, and total body skeletal muscle mass (TBMM) assessed using dual x-ray absorptiometry (DEXA). Next, we sought to examine if total myofibrillar and sarcoplasmic protein concentrations, myosin, and actin protein content, myofibrillar proteins involved with myofibril formation, or markers of mitochondrial physiology differed between clusters (high-response cluster (HI) = anabolic responders in the top 25th percentile and

low-response cluster (LO) = anabolic responders in the bottom 25th percentile). We hypothesized HI responders would exhibit greater changes in myofibrillar and sarcoplasmic protein concentrations relative to LO responders following RET. Additionally, we hypothesized HI responders would exhibit greater indices of mitochondrial volume or biogenesis relative to LO responders prior to and/or following RET.

## MATERIALS AND METHODS

### Ethical approval and study design

This study was approved by the Institutional Review Board at Auburn University (approved protocol #: 15-320 MR 1508) and conformed to the standards set by the latest revision of the Declaration of Helsinki. In the current study we analyzed muscle specimens from select participants that participated in a study we previously published (*Mobley et al., 2017*) and registered on ClinicalTrials.gov (Identifier: NCT03501628, date registered: April 18, 2018). However, the current study is not a clinical trial per the definition of the World Health Organization or National Institutes of Health given that health-related biomedical outcomes were not assessed. Apparently healthy, untrained college-aged male subjects provided written consent to participate in this study and performed a testing battery prior to (PRE) and 72 h after the last training bout (POST) following a 12-week full body RET program. The testing battery consisted of a VL muscle biopsy, full-body dual DEXA scan, and VL thickness assessment using ultrasound. More in-depth descriptions regarding the RET protocol as well as assessments of body composition, VL thickness, and fCSA can be found in past publications by our group (*Mobley et al., 2017*, *2018*).

### Muscle tissue processing

Muscle biopsies from PRE and POST testing sessions were collected using a five gauge needle under local anesthesia as previously described (*Mobley et al., 2017*). Immediately following tissue procurement, ~20–40 mg of tissue was embedded in cryomolds containing optimal cutting temperature media (Tissue-Tek®; Sakura Finetek Inc, Torrence, CA, USA) for fCSA assessment. The remaining tissue was teased of blood and connective tissue, wrapped in pre-labelled foils, flash frozen in liquid nitrogen ($LN_2$), and subsequently stored at $-80\ °C$ until protein and citrate synthase activity analyses described below.

### Western blotting of tissue lysates

For whole tissue lysate protein analysis, ~30 mg tissue was powdered on a $LN_2$-cooled ceramic mortar and placed in 1.7 mL microcentrifuge tubes on ice containing 500 μL of general cell lysis buffer (20 mM Tris–HCl (pH 7.5), 150 mM NaCl, 1 mM $Na_2$EDTA, 1 mM EGTA, 1% Triton; Cell Signaling, Danvers, MA, USA) pre-stocked with protease and Tyr/Ser/Thr phosphatase inhibitors (2.5 mM sodium pyrophosphate, 1 mM β-glycerophosphate, 1 mM $Na_3VO_4$, 1 μg/mL leupeptin). Samples were then homogenized on ice by hand using tight micropestles, insoluble proteins were removed with centrifugation at 500$g$ for 5 min, and obtained sample lysates were stored at $-80\ °C$ prior to Western blotting.
Upon first thaw total protein content was determined in duplicate using a BCA Protein Assay Kit (Thermo Scientific, Waltham, MA, USA). Lysates were immediately prepared thereafter for Western blotting using 4x Laemmli buffer at one μg/μL. Following sample preparation, 15 μL samples were loaded onto pre-casted gradient (4–15%) SDS-polyacrylamide gels (Bio-Rad Laboratories, Hercules, CA, USA) and subjected to electrophoresis (180 V for 45–60 min) using pre-made 1x SDS-PAGE running buffer (Ameresco, Framingham, MA, USA). Proteins were subsequently transferred (200 mA for 2 h) to polyvinylidene difluoride membranes (PVDF) (Bio-Rad Laboratories, Hercules, CA, USA), Ponceau stained, and imaged to ensure equal protein loading between lanes. Membranes were then blocked for 1 h at room temperature with 5% nonfat milk powder in Tris-buffered saline with 0.1% Tween-20 (TBST; Ameresco, Framingham, MA, USA). Rabbit anti-human Peroxisome Proliferator-Activated Receptor Gamma Coactivator 1-Alpha (PGC1-α, 1:1000; catalog #: GTX37356; GeneTex, Irvine, CA, USA), and mouse anti-human total OXPHOS antibody cocktail (1:250; catalog #:ab110413; Abcam, Cambridge, UK) were incubated with membranes overnight at 4 °C in TBST with 5% bovine serum albumin (BSA). The following day, membranes were incubated with horseradish peroxidase (HRP)-conjugated anti-rabbit IgG (catalog #: 7074; Cell Signaling, Danvers, MA, USA) or HRP-conjugated anti-mouse IgG (catalog #: 7072; Cell Signaling, Danvers, MA, USA) in TBST with 5% BSA at room temperature for 1 h (secondary antibodies diluted 1:2000). Membrane development was performed using an enhanced chemiluminescent reagent (Luminata Forte HRP substrate; EMD Millipore, Billerica, MA, USA), and band densitometry was performed using a gel documentation system and associated software (UVP, Upland, CA, USA). Raw densitometry values for each target were divided by whole-lane Ponceau densities, and these data were statistically analyzed between clusters. Regarding data presentation, values for a given protein target were normalized to the HI PRE group mean values whereby the HI PRE group average was 1.00, and data were expressed as relative expression units as reported in a recent publication by our laboratory (*Mobley et al., 2018*).

## Total myofibrillar and sarcoplasmic protein assessment

Myofibrillar and sarcoplasmic protein isolations were performed based on the methods of Goldberg's laboratory (*Cohen et al., 2009*). Briefly, frozen powdered muscle (8–13 mg) was weighed using an analytical scale sensitive to 0.0001 g (Mettler-Toledo, Columbus, OH, USA), and immediately placed in 1.7 mL polypropylene tubes containing 190 μL of ice cold homogenizing buffer (20 mM Tris–HCl, pH 7.2, 5 mM EGTA, 100 mM KCl, 1% Triton-X100). Samples were homogenized on ice using tight-fitting pestles, and centrifuged at 3000$g$ for 30 min at 4 °C. Supernatants (sarcoplasmic fraction) were collected, placed in 1.7 mL polypropylene tubes, and stored at −80 °C until concentration determination. The resultant pellet was resuspended in homogenizing buffer, and samples were centrifuged at 3000$g$ for 10 min at 4 °C. Resultant supernatants from this step were discarded, resultant pellets were resuspended in 190 μL of ice cold wash buffer (20 mM Tris–HCl, pH 7.2, 100 mM KCl, 1 mM DTT),

and samples were centrifuged at 3000$g$ for 10 min at 4 °C; this specific process was performed twice. Final myofibril pellets were resuspended in 200 μL of ice cold storage buffer (20 mM Tris–HCl, pH 7.2, 100 mM KCl, 20% glycerol, 1 mM DTT) and frozen at −80 °C until concentration determination.

Sarcoplasmic protein concentrations were determined in triplicate using the microplate BCA assay according to manufacturer's instructions (20 μL of 5x diluted sample + 180 μL Reagent A + B, absorbance reading at 580 nm) (Thermo Scientific, Waltham, MA, USA) and normalized to input muscle weights. The average coefficient of variation (CV) values for all triplicate readings were 1.6%. Myofibril protein concentrations were initially determined in triplicate using the microplate BCA assay (Thermo Scientific, Waltham, MA, USA). However, some wells (∼10%) contained noticeable myofibril aggregates yielding a relatively high average CV (9.2%). Hence, we adapted the BCA protocol to a cuvette-based assay whereby a larger volume of myofibril resuspensions were sampled (100 μL of 5x diluted sample + 900 μL Reagent A + B), and this visually yielded uniform absorbances in all samples. Samples were run in duplicate (not triplicate) using this method due to resource constraints, and the average CV proved to be lower for duplicate readings relative to the microplate method (5.0%). Myofibrillar protein concentrations from the cuvette-based assay were normalized to input muscle weights.

Proteins of select genes associated with new myofibril formation were assayed using aforementioned Western blotting techniques, but the myofibril fraction was assayed rather than the whole tissue lysate. For these assays, myofibril suspensions were prepared for and subjected to SDS-PAGE, proteins were transferred to PVDF membranes, membranes were Ponceau imaged, and membranes were blocked as described above. Rabbit anti-human ACTN2 (1:1000; catalog #: GTX103219; GeneTex, Irvine, CA, USA), rabbit anti-human SORBS2 (1:1000; catalog #: GTX81600; GeneTex, Irvine, CA, USA), rabbit anti-human MYOZ1 (1:1000; catalog #: GTX107334; GeneTex, Irvine, CA, USA), and rabbit anti-human MYOT (1:1000; catalog #: GTX109905; GeneTex, Irvine, CA, USA) were incubated with membranes overnight at 4 °C in TBST with 5% BSA. Thereafter, secondary antibody incubations, membrane development, and data procurement occurred similar to PGC1-α and OXPHOS described above.

### Determination of myosin heavy chain and actin content

SDS-PAGE preps from resuspended myofibrils were performed using: (a) 10 μL resuspended myofibrils, (b) 65 μL distilled water (diH$_2$O), and (c) 25 μL 4x Laemmli buffer. Samples were then loaded (15 μL) on pre-casted gradient (4–15%) SDS-polyacrylamide gels (Bio-Rad Laboratories, Hercules, CA, USA) and subjected to electrophoresis (200 V for 40 min) using pre-made 1x SDS-PAGE running buffer (Ameresco, Framingham, MA, USA). Following electrophoresis gels were rinsed in diH$_2$O for 15 min, and immersed in Coomassie stain (LabSafe GEL Blue; G-Biosciences, St. Louis, MO, USA) for 2 h. Thereafter, gels were destained in diH$_2$O for 60 min, bright field imaged using a gel documentation system (UVP), and band densities were determined using associated software. Myosin and actin content were expressed as arbitrary units/mg muscle.

## Sensitivity of BCA and Coomassie assessments

Given that our myofibril isolation protocol yielded an acceptable separation of the major contractile proteins from the sarcoplasmic fraction, we next examined the sensitivity of the BCA and Coomassie assays. For determination of myofibril BCA assay sensitivity, we performed an experiment where the following was loaded into cuvettes from the same subject (5x diluted sample + 900 μL Reagent A + B): 100 μL (low standard), 105 μL (5% greater protein content than low standard), 110 μL (10% greater protein content than low standard), 115 μL (15% greater protein content than low standard), 120 μL (20% greater protein content than low standard), and 125 μL (25% greater protein content than low standard). Following assay procedures, absorbance values were plotted against expected percent change values in protein content. For Coomassie stain sensitivity, we performed a similar experiment where the following was loaded onto a polyacrylamide gel from the same subject: 5.00 μL (low standard), 5.25 μL (5% greater protein content than low standard), 5.50 μL (10% greater protein content than low standard), 5.75 μL (15% greater protein content than low standard), 6.00 μL (20% greater protein content than low standard), and 6.25 μL (25% greater protein content than low standard). Following SDS-PAGE and Coomassie staining, densitometry values were assessed for myosin and actin and plotted against expected percent change values in myosin and actin content. Notably, lower volumes were used for this particular assay compared to what was loaded for actual actin and myosin analyses (15 μL) in HI and LO responders given sample volume limitations.

## Citrate synthase activity assays

Tissue lysates obtained through cell lysis buffer processing (described above) were batch processed for citrate synthase activity as previously described by our laboratory (Kephart et al., 2015). This metric was used as a surrogate for mitochondrial content per the findings of Larsen et al. (2012) suggesting citrate synthase activity highly correlates with transmission electron micrograph (TEM) images of mitochondrial content ($r = 0.84$, $p < 0.001$). The assay principle is based upon the reduction of 5,50-dithiobis(2-nitrobenzoic acid) (DTNB) at 412 nm (extinction coefficient 13.6 mmol/L/cm) coupled to the reduction of acetyl-CoA by the citrate synthase reaction in the presence of oxaloacetate. Briefly, five μg of skeletal muscle protein was added to a mixture composed of 0.125 mol/L Tris–HCl (pH 8.0), 0.03 mmol/L acetyl-CoA, and 0.1 mmol/L DTNB. The reaction was initiated by the addition of 5 μL of 50 mmol/L oxaloacetate and the absorbance change was recorded for 1 min. The average CV values for all duplicates was 4.6%.

## Responder clustering and statistical analysis

Similar to the methods of Davidsen et al. (2011), HI and LO anabolic responders were clustered based upon a summative score of PRE- to POST changes in multiple anabolic indices. Specifically, PRE- to POST changes in right leg VL muscle fCSA (type I + type II), VL thickness assessed via ultrasound, and TBMM assessed via DEXA were used, and each variable was equally weighted for the sum of scores.

Shapiro–Wilk tests of residuals for normality as well as Levene's Homogeneity of variance testing between clusters at PRE and POST were conducted for all dependent variables. Variables violating assumptions testing ($p \leq 0.050$) were square root-transformed for subsequent statistical testing. Assumptions testing results for all dependent variables are presented in Table 1. Briefly, actin content, ACTN2 protein levels, MYOT protein levels, complex II protein levels, complex III protein levels, and complex V protein levels were square root-transformed prior to statistical analyses due to Shapiro–Wilk $p$-values being <0.050 within a response cluster at PRE or POST. All dependent variable presented Levene test $p$-values > 0.050.

Dependent variable comparisons over time were analyzed between clusters using $2 \times 2$ (cluster $\times$ time) mixed factorial ANOVAs. If a significant cluster $\times$ time interaction was observed, PRE- to POST pairwise comparisons were performed within each cluster, and independent samples $t$-tests at the PRE and POST time points were performed to elucidate between-cluster differences. Significance was established at $p \leq 0.050$ for main effects and interactions, and $p \leq 0.025$ for post hoc tests (corrected for multiple comparisons). Post hoc Cohen's $d$ effect sizes were also calculated where significant main effects or interactions occurred, and effects were considered either small ($d \leq 0.500$), moderate ($d > 0.500$ and $d \leq 0.800$) or large ($d > 0.800$). Bivariate correlations were also performed on select variables, and significant correlations were established at $p \leq 0.050$. Assumptions tests, ANOVAs, and correlations were performed using SPSS v22.0 (IBM Corp, Armonk, NY, USA), and effect size calculations were performed using Microsoft Excel v2013 (Redmond, WA, USA).

# RESULTS

## Cluster differences in anabolic indices and other variables prior to and following training

Figure 1 diagrams DEXA TBMM, VL thickness, and fCSA values between clusters prior to and following training. HI responders ($n$ = 13) presented the following change scores (mean ± standard error values): $\Delta$TBMM = +3.1 ± 0.3 kg, $\Delta$VL thickness = +0.59 ± 0.05 cm, $\Delta$fCSA = +1,426 ± 253 $\mu m^2$. LO responders ($n$ = 12) presented the following change scores: $\Delta$TBMM = +1.1 ± 0.2, $\Delta$VL thickness = +0.24 ± 0.07, $\Delta$fCSA = +5 ± 209 $\mu m^2$. Notably, all $\Delta$ scores were significantly different when comparing HI versus LO responders ($p < 0.001$). Other dependent variables of interest that exhibited no between-group differences or cluster $\times$ time interactions included age (LO = 21 ± 1, HI = 21 ± 1; $p = 0.574$), PRE body mass (LO = 76.3 ± 3.8 years, HI = 74.1 ± 2.0 years; $p = 0.627$), total volume lifted during the RET program (LO = 303,973 ± 16,405 kg, HI = 323,805 ± 12,915 kg; $p = 0.348$), self-reported Caloric intake (LO PRE = 1,975 ± 140 kcal, LO POST = 2,376 ± 131 kcal, HI PRE = 1,953 ± 150 kcal, HI POST = 2,508 ± 193 kcal; cluster $p = 0.781$, time $p < 0.001$, cluster $\times$ time $p = 0.480$), and self-reported protein intake (LO PRE = 92 ± 6 g, LO POST = 137 ± 11 g, HI PRE = 90 ± 7 g, HI POST = 139 ± 12 g; cluster $p = 0.999$, time $p < 0.001$, cluster $\times$ time $p = 0.794$). Interestingly, and contrary to our previous report where only VL thickness was used as a clustering variable (*Mobley et al., 2018*), a significant cluster $\times$ time interaction was observed for three

| Table 1 Normality and homogeneity of variance test results of all dependent variables. | | | |
|---|---|---|---|
| **Variable** | **Shapiro–Wilk test (normality)** | **Levene's test (variance)** | **Data square root transformed?** |
| MF protein | HI PRE $p = 0.281$<br>HI POST $p = 0.718$<br>LO PRE $p = 0.587$<br>LO POST $p = 0.904$ | PRE $p = 0.457$<br>POST $p = 0.203$ | No |
| SARCO protein | HI PRE $p = 0.550$<br>HI POST $p = 0.304$<br>LO PRE $p = 0.261$<br>LO POST $p = 0.792$ | PRE $p = 0.830$<br>POST $p = 0.223$ | No |
| Myosin content | HI PRE $p = 0.829$<br>HI POST $p = 0.105$<br>LO PRE $p = 0.990$<br>LO POST $p = 0.074$ | PRE $p = 0.578$<br>POST $p = 0.831$ | No |
| Actin content | HI PRE $p = 0.483$<br>HI POST $p = 0.510$<br>LO PRE $p = 0.869$<br>**LO POST $p = 0.011$** | PRE $p = 0.578$<br>POST $p = 0.831$ | **Yes** |
| MF MYOZ1 protein | HI PRE $p = 0.964$<br>HI POST $p = 0.862$<br>LO PRE $p = 0.949$<br>LO POST $p = 0.410$ | PRE $p = 0.623$<br>POST $p = 0.753$ | No |
| MF SORBS2 protein | HI PRE $p = 0.357$<br>HI POST $p = 0.623$<br>LO PRE $p = 0.491$<br>LO POST $p = 0.876$ | PRE $p = 0.126$<br>POST $p = 0.548$ | No |
| MF ACTN2 protein | **HI PRE $p < 0.001$**<br>**HI POST $p = 0.002$**<br>LO PRE $p = 0.151$<br>LO POST $p = 0.768$ | PRE $p = 0.405$<br>POST $p = 0.115$ | **Yes** |
| MF MYOT protein | **HI PRE $p < 0.001$**<br>**HI POST $p = 0.010$**<br>LO PRE $p = 0.120$<br>LO POST $p = 0.700$ | PRE $p = 0.435$<br>POST $p = 0.104$ | **Yes** |
| CS activity | HI PRE $p = 0.288$<br>HI POST $p = 0.367$<br>LO PRE $p = 0.678$<br>LO POST $p = 0.502$ | PRE $p = 0.342$<br>POST $p = 0.095$ | No |
| PGC-1α protein | HI PRE $p = 0.193$<br>HI POST $p = 0.309$<br>LO PRE $p = 0.950$<br>LO POST $p = 0.699$ | PRE $p = 0.983$<br>POST $p = 0.351$ | No |
| Complex I protein | HI PRE $p = 0.116$<br>HI POST $p = 0.454$<br>LO PRE $p = 1.000$<br>LO POST $p = 0.963$ | PRE $p = 0.057$<br>POST $p = 0.561$ | No |
| Complex II protein | **HI PRE $p = 0.044$**<br>HI POST $p = 0.056$<br>LO PRE $p = 0.567$<br>LO POST $p = 0.270$ | PRE $p = 0.701$<br>PRE $p = 0.986$ | **Yes** |

| Table 1 (continued). | | | |
|---|---|---|---|
| **Variable** | **Shapiro–Wilk test (normality)** | **Levene's test (variance)** | **Data square root transformed?** |
| Complex III protein | HI PRE $p = 0.072$<br>**HI POST $p = 0.037$**<br>LO PRE $p = 0.089$<br>LO POST $p = 0.435$ | PRE $p = 0.599$<br>POST $p = 0.927$ | **Yes** |
| Complex IV protein | HI PRE $p = 0.055$<br>HI POST $p = 0.215$<br>LO PRE $p = 0.435$<br>LO POST $p = 0.082$ | PRE $p = 0.763$<br>POST $p = 0.729$ | No |
| Complex V protein | **HI PRE $p = 0.037$**<br>**HI POST $p = 0.011$**<br>LO PRE $p = 0.579$<br>LO POST $p = 0.074$ | PRE $p = 0.684$<br>POST $p = 0.706$ | **Yes** |

**Notes:**
All dependent variables were tested for normality and homogeneity of variance using Shapiro-Wilk and Levene's tests, respectively, and variables with $p$-values $\leq 0.050$ were square root transformed prior to statistical testing. HI, high-response cluster; LO, low-response cluster; PRE, muscle biopsy data at pre-training time point; POST, muscle biopsy following 12 weeks of resistance exercise training; all protein abbreviations can be found in the Methods section. Bold text indicates variables where Shapiro-Wilk test $p$-values were <0.05 and data were square root transformed prior to statistical analysis.

repetition (3RM) squat strength (LO PRE = 81 ± 6 kg, LO POST = 112 ± 5 kg, HI PRE = 77 ± 4 kg, HI POST = 120 ± 5 kg; cluster $p = 0.761$, time $p = 0.005$, cluster × time $p = 0.005$). Post hoc analyses indicated that both groups increased squat strength from PRE to POST ($p < 0.001$), although change in squat strength was greater in HI versus LO responders (42 ± 3 kg versus 31 ± 9 kg, respectively; $p = 0.005$) (refer to raw data in Supplemental File).

## Differences in myofibril and sarcoplasmic protein concentrations as well as myosin and actin content between clusters

No significant main effects or cluster × time interactions existed for myofibrillar protein concentrations (Fig. 2A), sarcoplasmic protein concentrations (Fig. 2B), or myofibrillar: sarcoplasmic protein ratios (Fig. 2C).

Additionally, no significant main effects or cluster × time interactions existed for myosin or actin content (Figs. 3A and 3B). However, there were significant positive associations between training-induced changes in myofibrillar protein concentrations versus myosin content ($r = 0.609$, $p = 0.001$; Fig. 3C) as well as changes in myofibrillar protein concentrations versus actin content ($r = 0.586$, $p = 0.002$; Fig. 3D). Notably, the positive associations between these variables provided us confidence that the cuvette-based myofibril assays obtained an accurate assessment of protein concentrations.

## Sensitivity of BCA and Coomassie assessments

For BCA assay sensitivity determination, increasing the volume of myofibril isolate from the same subject from 100 μL to 125 μL yielded positive increases in absorbance, and the line of best fit yielded an $r$-value of 0.927 (Fig. 4A). For myosin and actin band sensitivity determination, increasing the volume of myofibril isolate from the same subject from 5.00 μL to 6.25 μL also yielded positive increases in band densities, and the line of best fit for each marker yielded $r$-values above 0.950 (Figs. 4B and 4C). Thus, these results indicate these assays are capable of acutely detecting positive changes in protein content within ~±25%.

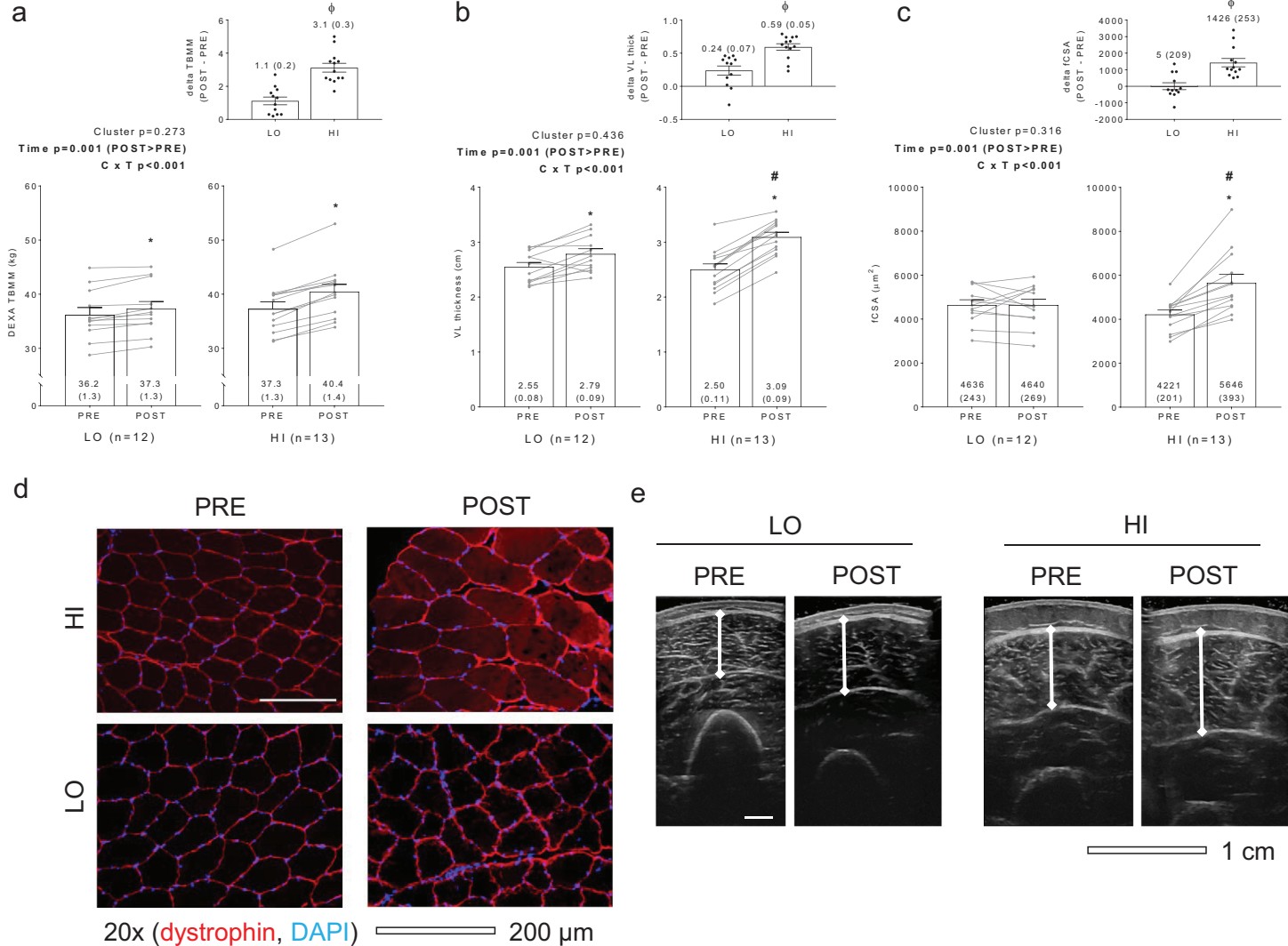

**Figure 1 Criterion variables for clustering.** PRE and POST values for total body muscle mass (TBMM) assessed via dual x-ray absorptiometry (DEXA) (A), vastus lateralis (VL) thickness assessed via ultrasound (B), and muscle fiber cross sectional area (fCSA) assessed via histology (C) between LO and HI response clusters. Notably, all three variables were used to develop HI and LO response clusters as outlined in the methods. All bar graph data are presented as means ± parenthesized standard error values, and bars are overlaid on individual responses. Additionally, POST—PRE values are presented as insets on (A–C). (D) contains representative fCSA and panel e contains representative VL thickness images of HI and LO responders. Study sites: *, POST > PRE within a given cluster ($p < 0.05$); #, HI POST > LO POST ($p < 0.05$); φ, delta (POST—PRE) value is greater in HI versus LO ($p < 0.001$).

## Differences in protein levels of genes involved with new myofibril formation between clusters

No significant main effects or cluster × time interactions existed for myofibrillar protein levels of ACTN2 (Fig. 5A), MYOT (Fig. 5B), or SORBS2 (Fig. 5D). A significant cluster × time interaction existed for MYOZ1 (Fig. 5C), and post hoc analysis indicated this marker increased in the LO cluster following training ($p = 0.025$). Additionally, delta MYOZ1 levels were significantly different between clusters ($p = 0.032$), and effect size

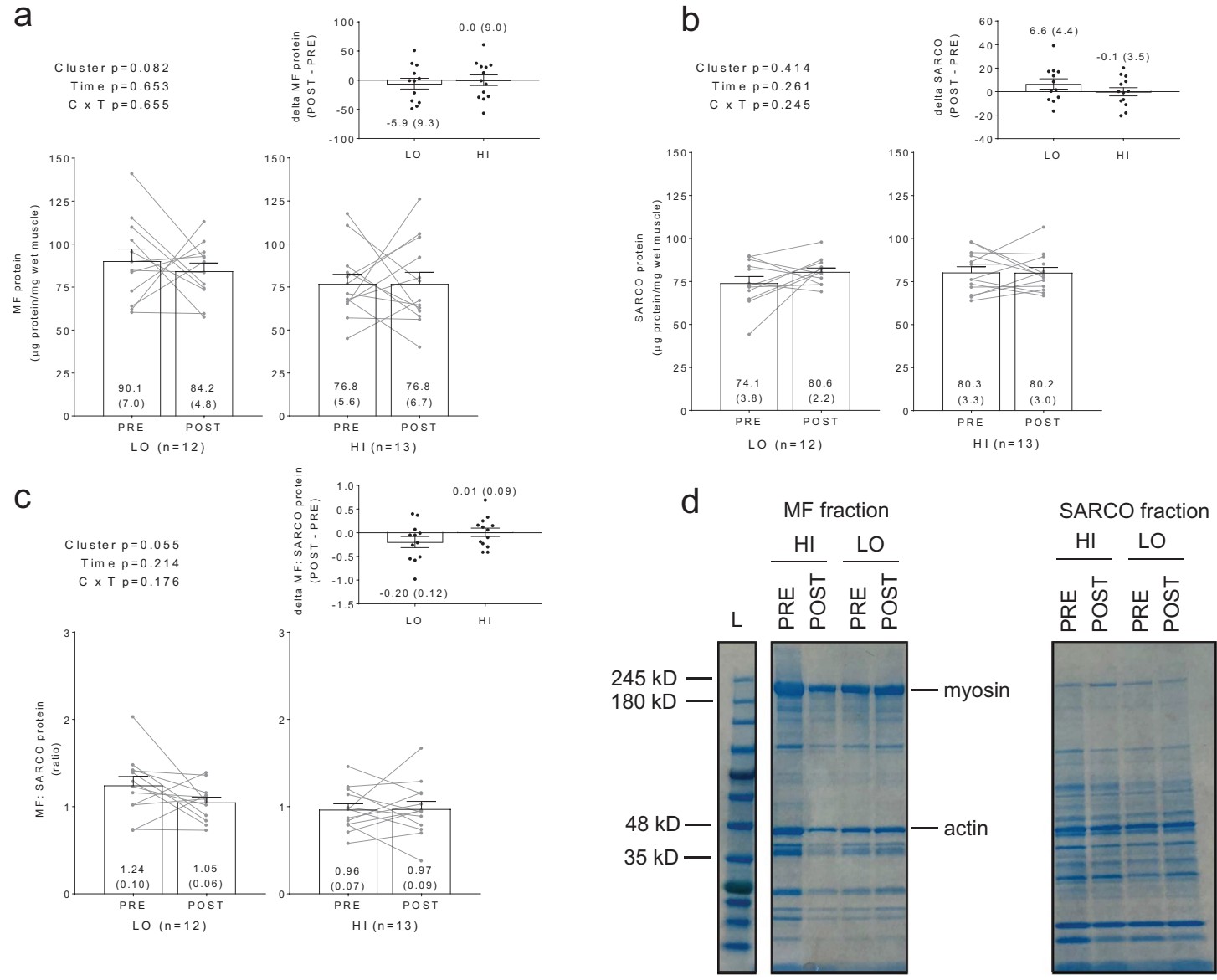

**Figure 2 Differences in myofibril and sarcoplasmic protein concentrations between clusters prior to and following training.** No significant main effects or cluster × time interactions existed for myofibrillar (MF) protein concentrations (A), sarcoplasmic (SARCO) protein concentrations (B), or MF: SARCO ratios (C). All bar graph data are presented as means ± parenthesized standard error values, and bars are overlaid on individual responses. Additionally, POST—PRE values are presented as insets on (A–C). (D) is a representative image from HI and LO subjects demonstrating a Coomassie stain of isolated MF and SARCO proteins.

calculations indicated a moderate effect existed regarding the up-regulation of MYOZ1 in the LO cluster (Cohen's $d$ = 0.691).

## Associations between changes in myofibrillar and sarcoplasmic protein concentrations with other dependent variables

When all subjects were pooled for analysis, no significant association existed between delta myofibrillar protein concentrations and delta fCSA levels ($r$ = −0.014, $p$ = 0.947; Fig. 6A).

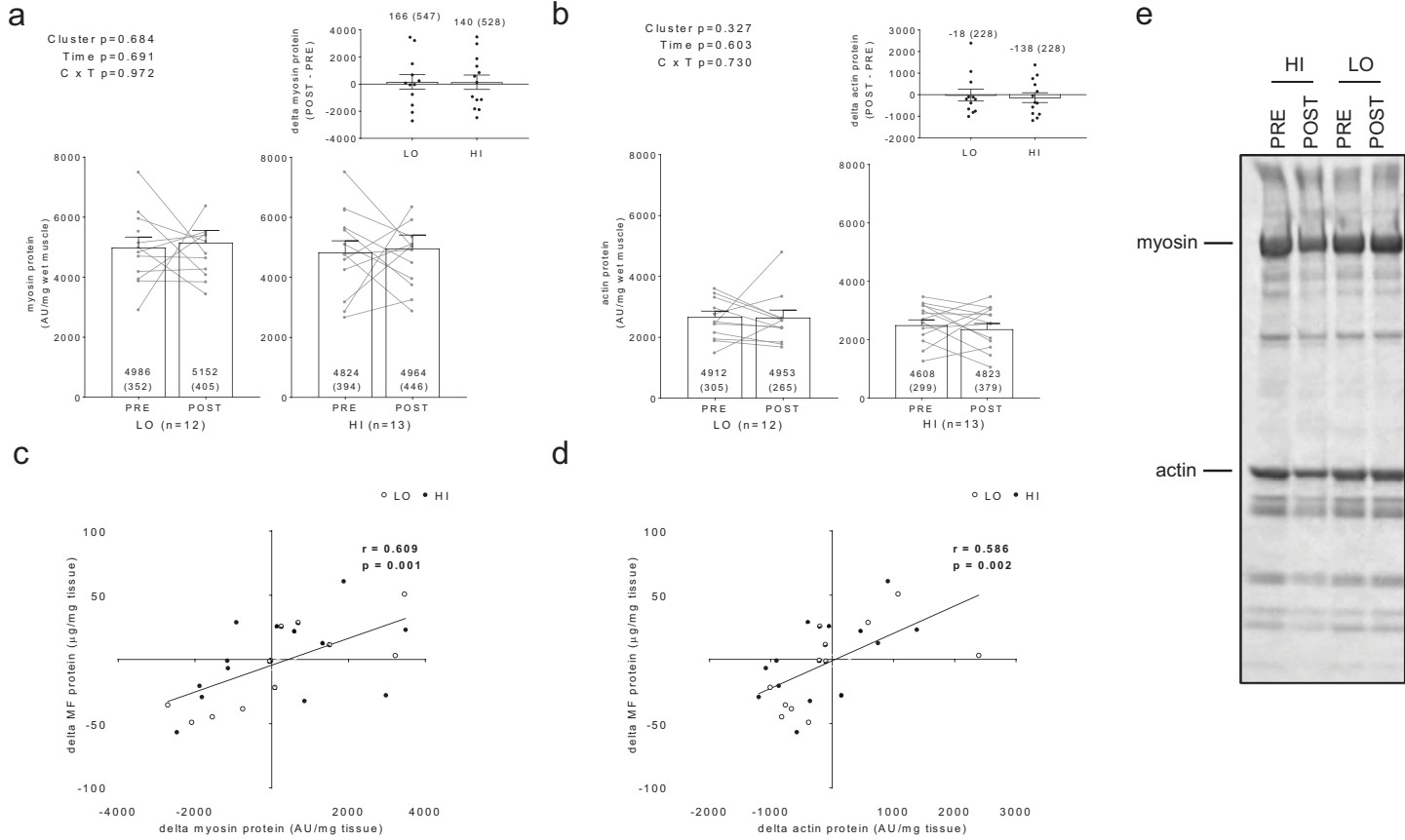

**Figure 3** **Differences in myosin and actin content between clusters prior to and following training.** No significant main effects or cluster × time interactions existed for myosin content (A) or actin content (B). All bar graph data are presented as means ± parenthesized standard error values, and bars are overlaid on individual responses. Additionally, POST—PRE values are presented as insets on (A and B). There were significant positive associations between change in MF protein versus change in myosin content (C) and actin content (D). (E) is a representative image from HI and LO responders demonstrating a Coomassie stain of MF proteins containing myosin and actin (converted to grayscale for band densitometry analysis).

Significant negative associations existed between delta myofibrillar protein concentrations and PRE fCSA ($r = -0.467$, $p = 0.019$; Fig. 6B) as well as PRE myofibrillar protein concentrations ($r = -0.758$, $p < 0.001$; Fig. 6C). No significant associations existed between delta sarcoplasmic protein concentrations and delta fCSA ($r = 0.091$, $p = 0.666$; Fig. 5D) or PRE fCSA ($r = -0.113$, $p = 0.591$; Fig. 6E). A significant negative association existed between delta sarcoplasmic concentrations and PRE sarcoplasmic concentrations ($r = -0.763$, $p < 0.001$; Fig. 6F).

## Differences in mitochondrial volume and biogenesis markers between clusters

No significant main effects or cluster × time interactions existed for PGC-1α protein levels (Fig. 7A). Interestingly, there were significant main effects of group (HI > LO, $p = 0.018$) and time (PRE > POST, $p = 0.037$) for citrate synthase activity (Fig. 7B); however, no cluster × time interaction existed ($p = 0.612$). Regarding the main effect of time, effect size calculations indicated a moderate effect existed regarding

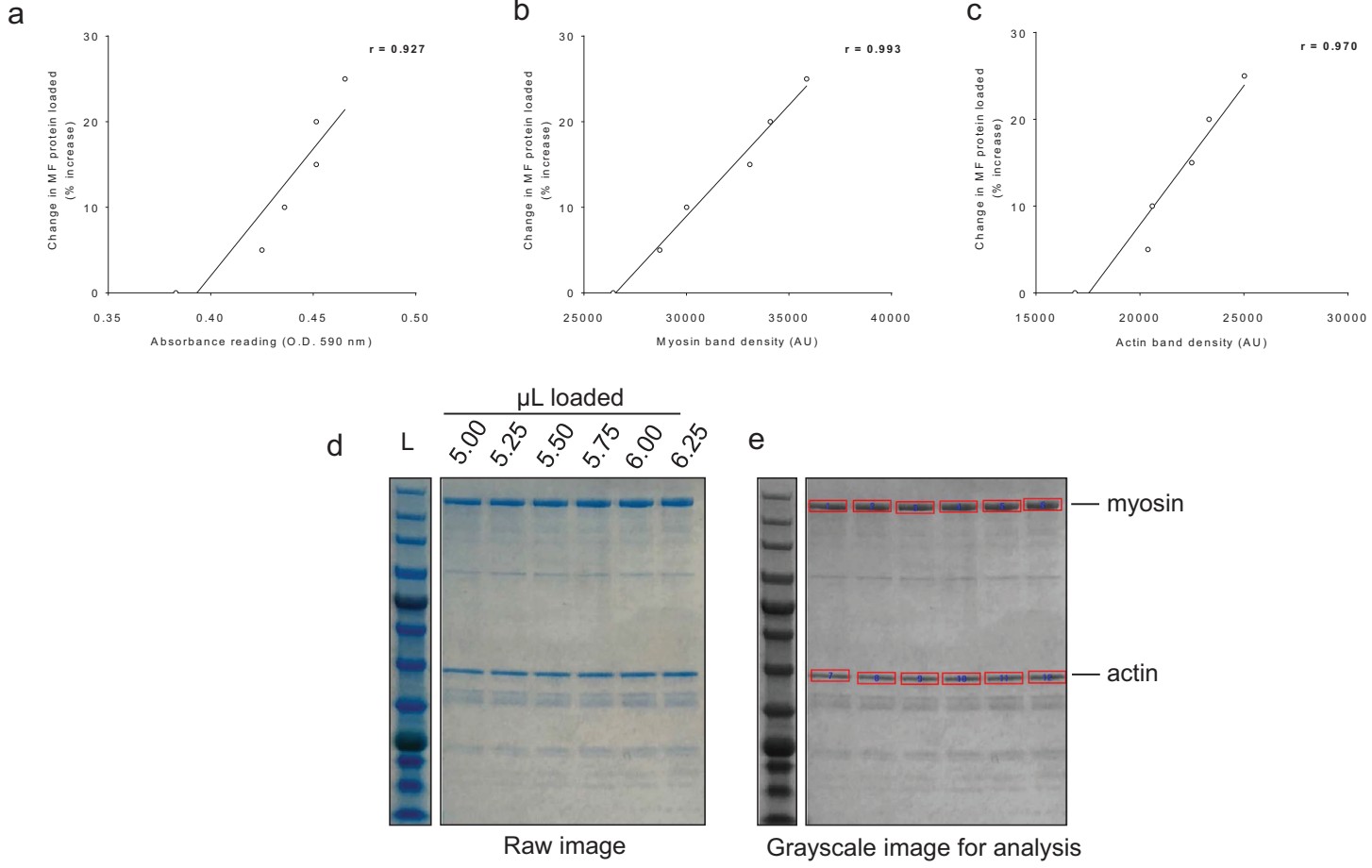

**Figure 4 Sensitivity of BCA and Coomassie stain for detecting protein changes.** (A) depicts BCA assay data of increasing myofibril isolate (*y*-axis) plotted against absorbance readings (*x*-axis). (B and C) depict increasing myofibril isolate (*y*-axes) plotted against myosin and actin band densitometries, respectively (*x*-axes). Collectively, these data indicate that small changes in protein content (i.e., within 25% changes) can be accurately assessed using these assays. Experimental details of these sensitivity assessments are provided in the methods. (D) illustrates a digital image captured for the myosin and actin sensitivity assays. (E) depicts the raw image converted to grayscale with red rectangles being a software tool feature used to extrapolate total band density.

the down-regulation of citrate synthase activity due to training ($d = -0.589$). Regarding the main effect of cluster, effect size calculations indicated a moderate effect existed regarding citrate synthase activity being greater in HI versus LO responders ($d = 0.737$). There were no significant main effects or cluster $\times$ time interactions for complex I protein levels (cluster $p = 0.834$, time $p = 0.097$, cluster $\times$ time $p = 0.644$), complex II protein levels (cluster $p = 0.807$, time $p = 0.761$, cluster $\times$ time $p = 0.737$), complex III protein levels (cluster $p = 0.836$, time $p = 0.561$, cluster $\times$ time $p = 0.479$), complex IV protein levels (cluster $p = 0.885$, time $p = 0.502$, cluster $\times$ time $p = 0.810$), or complex V protein levels (cluster $p = 0.782$, time $p = 0.506$, cluster $\times$ time $p = 0.608$) (Fig. 7C). Notably, only $n = 12$ of 13 HI responders were assayed for all targets presented in Fig. 7 due to the lack of lysate volume for one subject in this cluster.

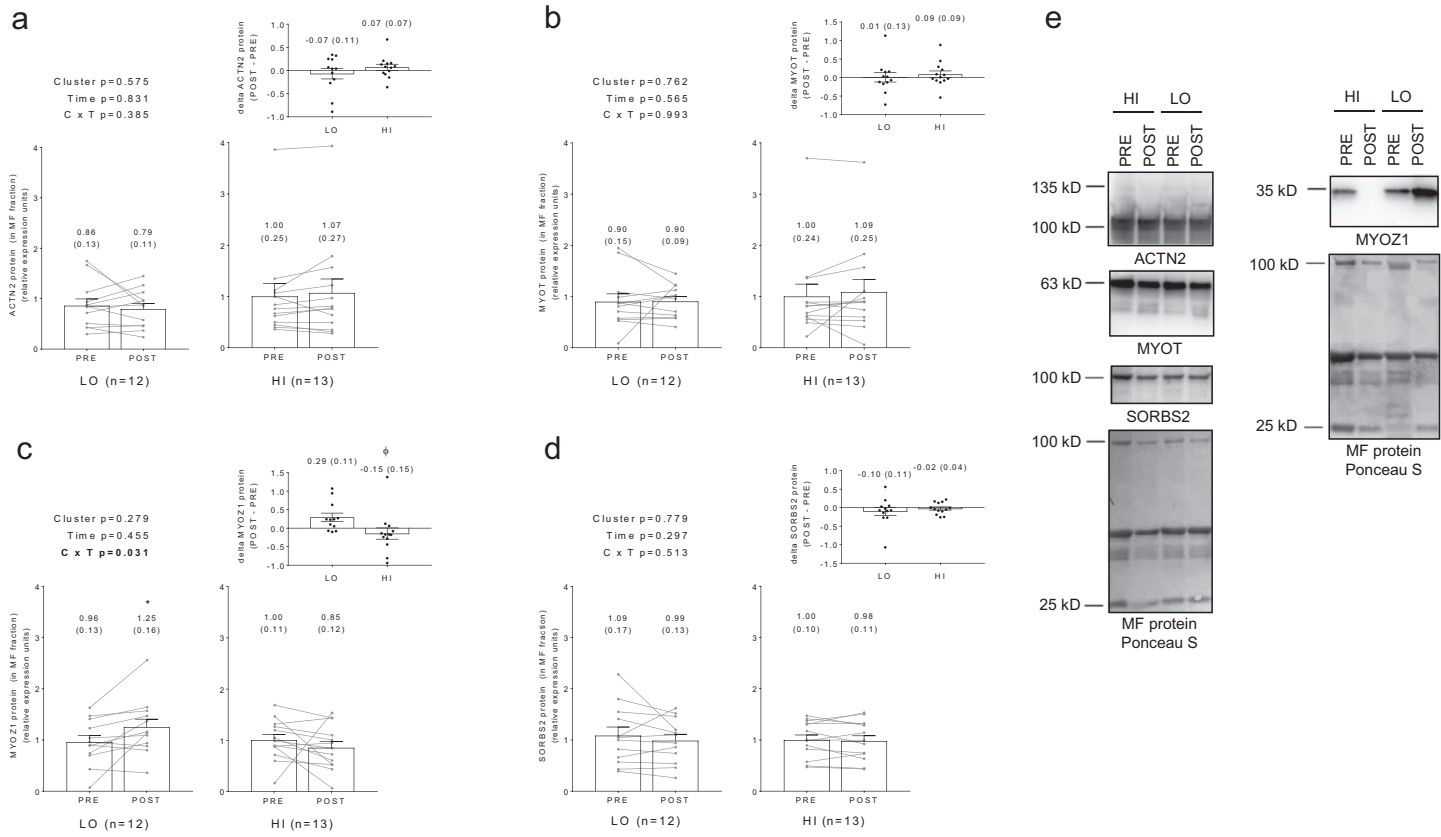

**Figure 5 No significant main effects or cluster × time interactions existed for myofibrillar protein levels of ACTN2 (A), MYOT (B), or SORBS2 (D).** A significant cluster × time interaction existed for MYOZ1 (C), and post hoc analysis indicated this marker increased in the LO cluster following training (*, $p = 0.025$). Additionally, delta MYOZ1 levels were significantly different between clusters (φ, $p = 0.032$). All bar graph data are presented as means ± parenthesized standard error values, and bars are overlaid on individual responses. Additionally, POST—PRE values are presented as insets (A–D). (E) depicts representative Western blot images for each of the presented protein markers in this figure.

## DISCUSSION

There is a general consensus RET-induced increases in muscle fCSA likely coincide with increased myofibrillar protein. Historical studies performed in rodents and small animals suggest increased levels of physical activity (*Helander, 1961*) or mechanical overload (*Goldberg, 1968*; *Goldspink, 1964*) increase myofibril protein density. Additionally, one of these studies correlated an increase in myofibril number per fiber to an increased muscle fiber size in mice subjected to RET (*Goldspink, 1964*). However, other studies in small animals also suggest mechanical overload decreases (*Seiden, 1976*) or does not alter (*Goldspink & Howells, 1974*) myofibril protein density. Given the increased research emphasis in exercise science examining signaling phenomena (e.g., phosphorylation patterns), genomic and transcriptomic events, and tracer kinetics involved with protein turnover, little evidence exists in humans regarding how chronic RET affects changes in myofibrillar or sarcoplasmic protein concentrations. In this regard, while RET generally increases fCSA, limited evidence suggests training may decrease myofibril protein density while potentially increasing sarcoplasmic protein density (*Luthi et al., 1986*;

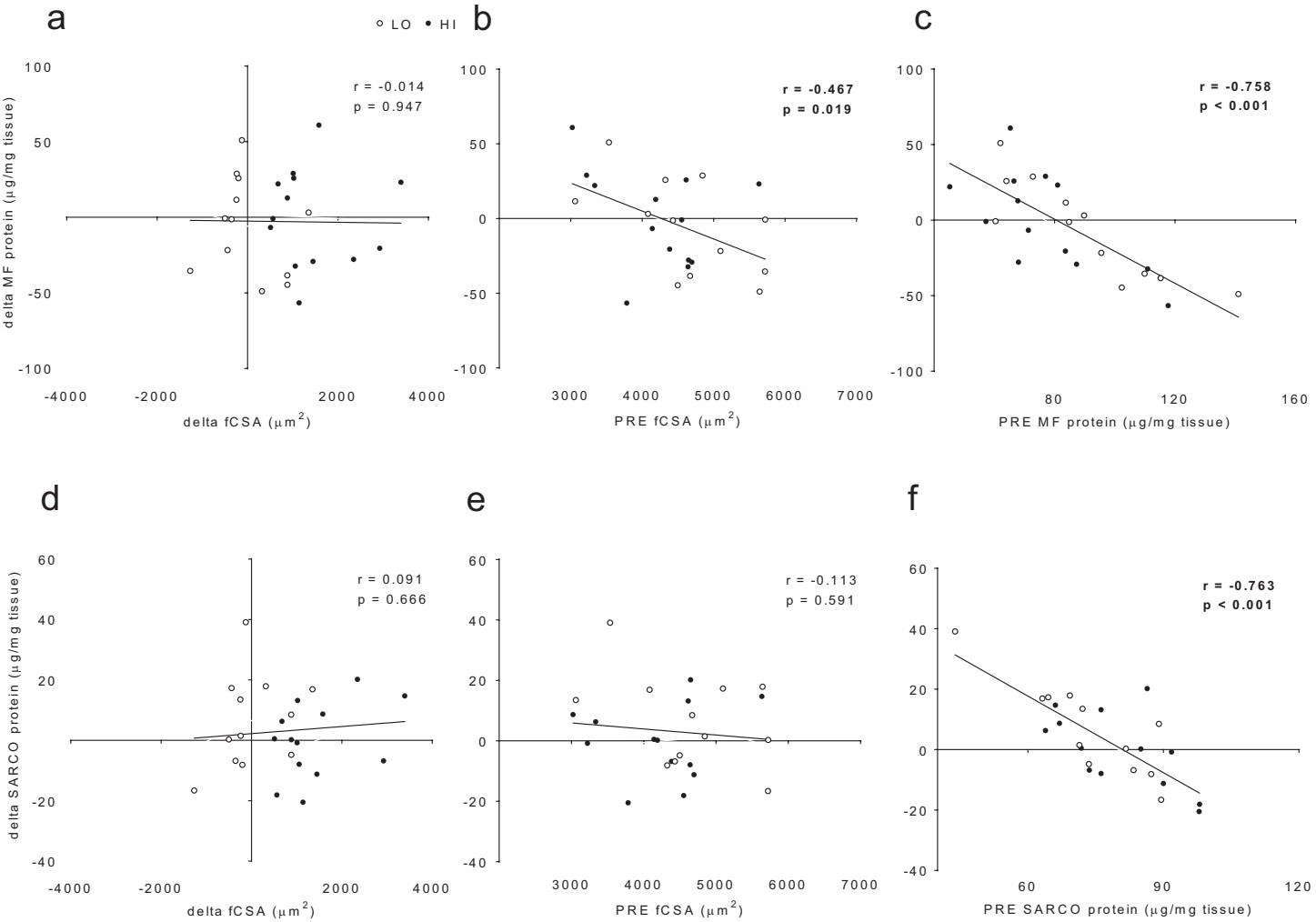

**Figure 6 No significant association existed between delta (POST—PRE) myofibrillar (MF) protein concentrations and delta muscle fiber cross sectional area (fCSA) levels (A).** Significant negative associations existed between delta myofibrillar (MF) protein concentrations and PRE fCSA (B) as well as PRE MF protein concentrations (C). No significant associations existed between delta sarcoplasmic (SARCO) protein concentrations and delta fCSA (D) or PRE fCSA (E). A significant negative association existed between delta SARCO concentrations and PRE SARCO concentrations (F).

*MacDougall et al., 1982*; *Toth et al., 2012*). We observed total myofibrillar and sarcoplasmic protein concentrations (i.e., densities) as well as total myosin and actin content were not differentially altered between or within HI and LO responders following 12 weeks of RET. These data suggest fCSA increases in HI cluster subjects during the short-term RET employed herein may be more related to factors other than myofibrillar or sarcoplasmic protein concentration increases (e.g., a potential accumulation of intracellular fluid). While this is a bold hypothesis, there is limited evidence in the literature to suggest cellular fluid shifts may account for RET-induced increases in fCSA. Specifically, *Kadi et al. (2004)* examined periodic fCSA changes in college-aged men following 90 days of RET as well as 90 days following detraining. Interestingly, the authors reported that fCSA increased ~16% from pre-training levels following the last training

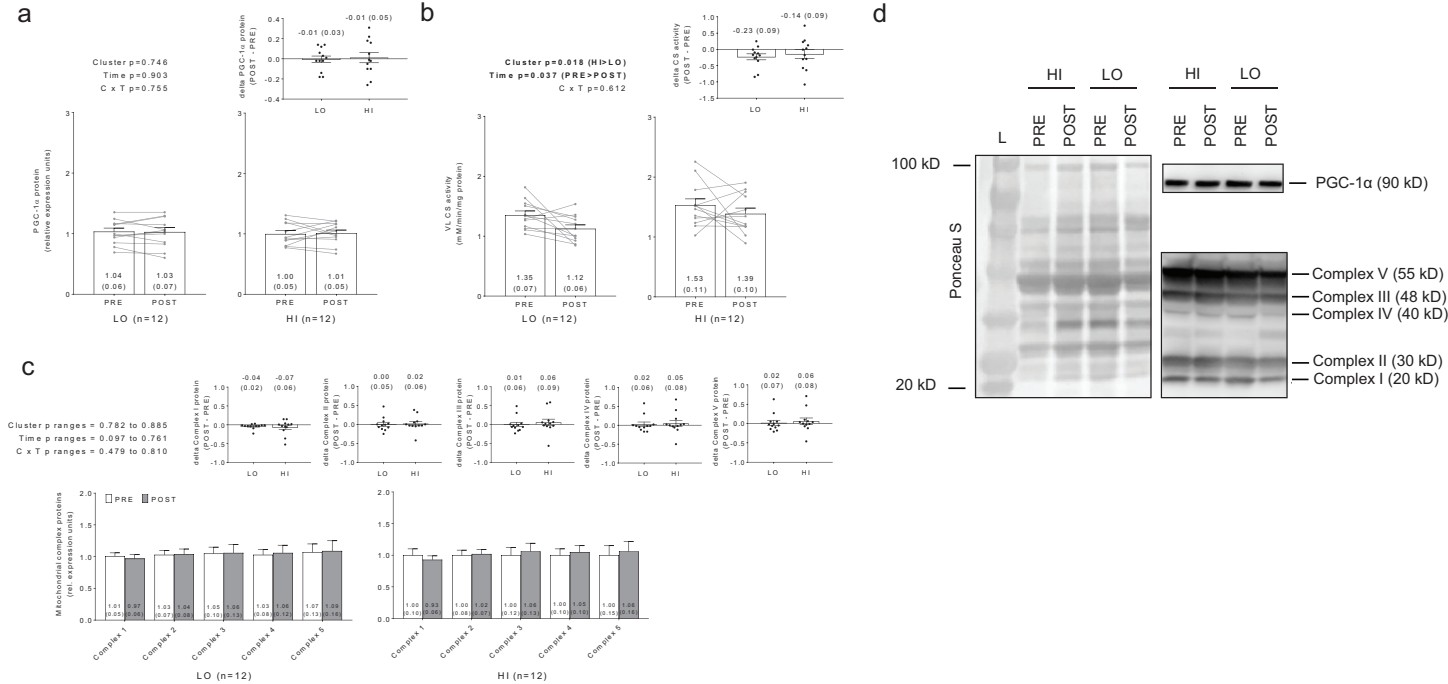

**Figure 7 No significant main effects or cluster × time interaction existed for PGC-1α protein levels (A).** Main effects of cluster and time existed for citrate synthase activity levels, although no cluster × time interaction existed (B). No significant main effects or cluster × time interaction existed for mitochondrial complex I–V protein levels (C). All bar graph data are presented as means ± parenthesized standard error values, and bars are overlaid on individual responses except for (C). Additionally, POST—PRE values are presented as insets (A–C). (D) depicts representative Western blot images for each of the presented protein markers in this figure.

bout, whereas fCSA levels returned to pre-training levels only after 10 days of detraining and were maintained thereafter. While reasons for these observations were not speculated by the authors, we posit that the latter phenomena was due to either: (a) MyoPS rates robustly decreasing within a 10-day time frame following training in lieu of elevated muscle protein breakdown rates (i.e., a net negative protein balance within the first 10 days of detraining), (b) muscle protein breakdown rates robustly and abnormally increasing within a 10-day time frame following training which, again, resulted in a net negative protein balance within the first 10 days of detraining, or (c) a restoration of intracellular fluid levels to pre-training values. From our perspective, there is no current evidence in the scientific literature and/or physiological rationale to support the former two explanations. Thus, future research examining the contribution of intracellular fluid shifts to RET-induced muscle fiber hypertrophy is warranted.

The strong negative associations between pre-training myofibrillar and sarcoplasmic protein concentrations versus RET-induced changes in these variables are compelling. In essence, we observed individuals with lower pre-training myofibrillar or sarcoplasmic protein concentrations presented a greater propensity for protein accretion whereas those with higher pre-training levels experienced a dilution effect (or even loss) in protein following RET, and these phenomena seemingly occurred regardless of cluster. These data also suggest a threshold may exist in relation to intracellular protein accretion

given that individuals with high PRE myofibrillar or sarcoplasmic protein concentrations (e.g., > ~90 μg/mg) did not experience POST increases in protein levels. The concept that a cellular protein accretion threshold exists is not novel. For instance, *Soltani et al. (2016)* recently posited proliferating cells reach a protein accretion threshold prior to undergoing cell division, and this threshold may trigger cell division to occur. However, since skeletal muscle fibers are post-mitotic, fibers which reach a theoretical myofibrillar or sarcoplasmic concentration threshold during RET could down-regulate synthesis or up-regulate proteolysis in order to prevent further accretion from occurring. While this is a provocative explanation of our observations, we do lack mechanistic data to support this hypothesis and future research in this area is needed.

Since muscle hypertrophy is generally viewed as energetically costly, we also sought to determine if mitochondrial markers were differentially expressed between high versus low anabolic responders to RET. We hypothesized high anabolic responders may experience greater increases in markers related to mitochondrial volume with RET or possess greater pre-training values of these markers to support the energetic demands of increased anabolism relative to low responders. However, contrary to our hypothesis, mitochondrial biogenesis or greater mitochondrial involvement with muscle hypertrophy in HI responders is seemingly paradoxical given that the intramuscular signaling mechanisms which facilitate muscle protein accretion with RET and mitochondrial biogenesis with endurance training are dichotomous (*Atherton et al., 2005*). We observed RET decreases whole-tissue citrate synthase activity regardless of cluster, although levels were greater in HI versus LO responders throughout the intervention. Additionally, all other assayed markers (i.e., mitochondrial complex and PGC-1α protein levels) were not differentially expressed between the LO versus HI clusters prior to or following training. These findings suggest: (a) skeletal muscle mitochondrial volume does not change (in accordance with no change observed in complex protein levels) or may decrease in response to the prescribed RET (in accordance with decreased citrate synthase activity levels) regardless of response cluster, and (b) the HI cluster may have expressed a greater mitochondrial volume relative to the LO cluster throughout the intervention (in accordance with greater citrate synthase activity levels). The former hypothesis is supported by numerous studies which report one to six months of RET may decrease mitochondrial volume through a possible dilution effect (i.e., no true change in mitochondrial content combined with increased fiber size) (*Chilibeck, Syrotuik & Bell, 1999*; *Kon et al., 2014*; *Luthi et al., 1986*; *MacDougall et al., 1979*), although recent evidence suggests that mitochondrial function (i.e., complex I-supported respiration) can be enhanced through 12 weeks of RET (*Porter et al., 2015*). However, the latter hypothesis is a novel in that no study has suggested greater mitochondrial volume promotes increases in RET-induced hypertrophy. Evidence outside of the exercise science literature has associated mitochondrial involvement with cell size regulation. For instance, cachexia-induced muscle fiber atrophy is commonly associated with decreased mitochondrial biogenesis and function (*VanderVeen, Fix & Carson, 2017*). Additionally, in vitro studies have established greater mitochondrial volume or increased mitochondrial function promote increases in cell growth through enhanced transcription and

translation (*Miettinen & Bjorklund, 2017*). Some evidence in the exercise science literature also hints at mitochondrial involvement for facilitating skeletal muscle hypertrophy. For instance, a recent review by *Murach & Bagley (2016)* cites three studies (*Lundberg et al., 2013*; *Lundberg, Fernandez-Gonzalo & Tesch, 2014*; *Mikkola et al., 2012*) which demonstrated that concurrent RET with endurance training facilitates greater skeletal muscle hypertrophy compared to RET alone. While not explicitly stated in the review, these findings do provide rationale for endurance exercise-induced increases in mitochondrial biogenesis potentially supporting RET-induced increases in muscle growth. It should be noted that the current study is limited due to markers of mitochondrial function (e.g., state II–IV respiration and complex activities) not being assayed. Additionally, greater citrate synthase activity in HI versus LO responders could be due to other factors independent of mitochondrial volume changes (e.g., greater citrate synthase production in HI responders in the absence of mitochondrial volume expansion, or a greater turnover of this particular enzyme in LO responders). Notwithstanding, our data suggest mitochondrial volume may be involved with RET-induced hypertrophy, and this relationship should be further examined.

Increased myofibrillar MYOZ1 protein levels in the LO cluster following RET is an interesting observation. Specifically, levels appreciably increased in seven LO responders following training, whereas only one HI responder showed a robust elevation in this target. Beyond being a biomarker of new myofibril formation, MYOZ1 has been reported to be localized to the Z-disc in mature skeletal and cardiac muscle fibers where it acts to abrogate calcineurin signaling (*Frey et al., 2008*). Interestingly, MYOZ1-knockout mice present increased running performance when provided voluntary running wheels (i.e., nightly distance and running pace), increased type I muscle fCSA, and a resilience in muscle atrophy in response to cardio-toxin-induced injury (*Frey et al., 2008*). Additionally, synergist ablation in mice induces a robust down-regulation (~30–66%) in MyoZ1 mRNA levels 1–14 days following surgery in the presence of robust plantaris hypertrophy (*Chaillou et al., 2013*). Based upon this collective evidence, it is plausible that an up-regulation in myofibrillar MYOZ1 protein levels in low anabolic responders may mechanistically interfere with muscle hypertrophy through some unresolved signaling mechanism. However, our findings are limited given that our *n*-sizes per cluster group were relatively small, and no mechanistic validation of this target affecting muscle cell size was performed herein. Therefore, future mechanistic studies examining the role that MYOZ1 protein plays in RET-induced training adaptations are warranted.

What should finally be noted is the manner in which anabolic response clusters are generated. Previously, we implemented a K-means cluster analysis based solely upon changes in VL thickness changes to generate low, moderate, and high anabolic response clusters, and completely separate dependent molecular variables were analyzed (*Mobley et al., 2018*). Our prior approach is similar to several reports which have used K-means cluster analysis in total or type II fCSA changes as clustering variable (*Bamman et al., 2007*; *Kim et al., 2007*; *Petrella et al., 2008*; *Stec et al., 2016*; *Thalacker-Mercer et al., 2013*; *Thalacker-Mercer, Petrella & Bamman, 2009*). More recent data from *Davidsen et al. (2011)* used a combination of metrics (e.g., fCSA,

whole-body lean tissue mass changes, strength gains for three leg exercises) and selected subjects in the upper and lower 15–25th percentiles of these metrics to define high and low anabolic responders. We adopted a similar approach herein by generating clusters based upon DEXA TBMM, VL thickness, and fCSA, and selecting subjects in the upper and lower quartiles. Notably, in our past publication we did not observe a between-cluster interaction for 3RM squat strength, although our newly-adopted approach yielded a cluster × time interaction for 3RM squat strength whereby strength gains were greater in the HI responders (HI Δ3RM squat = 42 ± 3 kg, LO Δ3RM squat = 31 ± 9 kg, respectively; $p = 0.005$). Additionally, we previously noted that ribosome density increased more so in HI versus LO responders following RET (*Mobley et al., 2018*), and our new clustering approach herein yielded similar results for ribosome density (HI Δtotal RNA = 113.6 ± 29.4 ng/mg muscle, LO Δtotal RNA = 30.3 ± 30.6 ng/mg muscle, respectively; $p = 0.062$, to raw data in Supplemental File). In line with our previous data, the current multifactorial clustering method yielded significant main effect of time increases, but no between-cluster differences satellite cell changes (cluster × time $p = 0.472$, time $p < 0.001$; to raw data in Supplemental File). However, it is interesting that the current clustering method indicated HI responders experienced significantly greater increases in type I fiber myonuclear accretion with training (LO PRE = 2.8 ± 0.1 nuclei/fiber, LO POST = 3.1 ± 0.3 nuclei/fiber, HI PRE = 2.2 ± 0.2 nuclei/fiber, HI POST = 3.3 ± 0.3 nuclei per fiber; cluster × time $p = 0.048$, time $p = 0.002$) as well as trended towards increasing type II fiber myonuclear accretion with training (LO PRE = 3.0 ± 0.2 nuclei/fiber, LO POST = 3.8 ± 0.3 nuclei/fiber, HI PRE = 2.7 ± 0.3 nuclei/fiber, HI POST = 4.1 ± 0.3 nuclei per fiber; cluster × time $p = 0.048$, time $p = 0.002$) (to raw data in Supplemental File). Importantly, given that the combined approach of anabolic indices yielded differences in a functional strength outcome as well as myonuclear accretion differences, the latter which has been posited to contribute to higher hypertrophic responses to RET (*Petrella et al., 2008*), we feel using multiple metrics to define anabolic responders (e.g., fCSA + DEXA data + limb circumference + muscle thickness, etc.) could be a more insightful methodology for those publishing on the topic moving forward.

## CONCLUSIONS

The current data suggest myofibrillar or sarcoplasmic protein accretion differences do not exist between high versus low anabolic responders to prior to or following a 12-week RET. We interpret these findings to suggest myofibrillar or sarcoplasmic protein concentration alterations during a shorter-term RET program assume a minor role in hypertrophic magnitudes in previously untrained college-aged subjects. However, longer-term interventions (i.e., six+ months) may yield different outcomes. Training decreased whole-tissue lysate citrate synthase activity regardless of cluster, although levels were greater in HI versus LO responders throughout RET. We interpret these findings to suggest that individuals possessing more mitochondria throughout an RET program may experience greater muscle hypertrophy. However, citrate synthase is only one surrogate marker of whole-tissue mitochondrial volume and future studies examining how other

markers of volume (e.g., TEM micrographs) as well as mitochondrial function are affected between HI versus LO responders will better validate this hypothesis. Finally, the current data combined with prior rodent studies suggest an up-regulation in myofibril MYOZ1 protein levels may interfere with muscle hypertrophy, and this novel target should be further investigated at a mechanistic level.

## ACKNOWLEDGEMENTS

We thank our participants for devoting time to the study, as well as our training staff including Jessica Nendze, Casey Sexton, Chester Davidson, Romil Patel, David Baumohl and Corey Chandler for their contributions.

### Funding

Participant compensation costs were provided through a gift to Michael D. Roberts from Hilmar Ingredients and BioNutritional Research Group. Michael D. Roberts and Andreas N. Kavazis used discretionary laboratory funds for reagent costs. The funders had no role in study design, data collection and analysis, decision to publish, or preparation of the manuscript.

### Grant Disclosures

The following grant information was disclosed by the authors:
Hilmar Ingredients and BioNutritional Research Group.

### Competing Interests

Christopher Lockwood is the CEO of Lockwood, LCC where he consults with nutritional supplement companies and mainstream media outlets. He has no competing interests in publishing these data. All other authors declare that they have no competing interests.

### Author Contributions

- Michael D. Roberts conceived and designed the experiments, performed the experiments, analyzed the data, contributed reagents/materials/analysis tools, prepared figures and/or tables, authored or reviewed drafts of the paper, approved the final draft.
- Matthew A. Romero performed the experiments, analyzed the data, authored or reviewed drafts of the paper, approved the final draft.
- Christopher B. Mobley performed the experiments, analyzed the data, authored or reviewed drafts of the paper, approved the final draft.
- Petey W. Mumford performed the experiments, analyzed the data, authored or reviewed drafts of the paper, approved the final draft.
- Paul A. Roberson performed the experiments, analyzed the data, authored or reviewed drafts of the paper, approved the final draft.
- Cody T. Haun performed the experiments, analyzed the data, authored or reviewed drafts of the paper, approved the final draft.

- Christopher G. Vann performed the experiments, analyzed the data, authored or reviewed drafts of the paper, approved the final draft.
- Shelby C. Osburn performed the experiments, analyzed the data, authored or reviewed drafts of the paper, approved the final draft.
- Hudson H. Holmes performed the experiments, analyzed the data, authored or reviewed drafts of the paper, approved the final draft.
- Rory A. Greer performed the experiments, analyzed the data, authored or reviewed drafts of the paper, approved the final draft.
- Christopher M. Lockwood conceived and designed the experiments, analyzed the data, contributed reagents/materials/analysis tools, authored or reviewed drafts of the paper, approved the final draft.
- Hailey A. Parry performed the experiments, analyzed the data, authored or reviewed drafts of the paper, approved the final draft.
- Andreas N. Kavazis performed the experiments, analyzed the data, authored or reviewed drafts of the paper, approved the final draft.

## Human Ethics

The following information was supplied relating to ethical approvals (i.e., approving body and any reference numbers):

This study was approved by the Institutional Review Board at Auburn University (approved protocol #: 15-320 MR 1508).

## Data Availability

The raw data are provided in a Supplemental File.

## Supplemental Information

Supplemental information for this article can be found online at http://dx.doi.org/10.7717/peerj.5338#supplemental-information.

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
