# Peer review of "Skeletal muscle mitochondrial volume and myozenin-1 protein differences exist between high versus low anabolic responders to resistance training"

_PeerJ, doi:10.7717/peerj.5338_

## Round 0.1 · original submission · Minor Revisions

The three reviewers and I find many positives about the study and the quality of the written manuscript and therefore we are happy for you to resubmit your manuscript to address some of the generally minor comments of the reviewers.

·

Basic reporting

Please find my review within the enclosed document.

Experimental design

Please find my review within the enclosed document.

Validity of the findings

Please find my review within the enclosed document.

Comments for the author

Please find my review within the enclosed document.

·

Basic reporting

No comment

Experimental design

No comment

Validity of the findings

No comment

Comments for the author

This paper by Robert’s and co-workers investigated whether total myofibrillar and sarcoplasmic protein concentrations, myosin and actin protein content, myofibrillar protein levels of genes involved with myofibril formation, or markers of mitochondrial physiology present differences in expression between ‘high’ and ‘low’ responder clusters to a 12-week resistance training program. In contrast to their hypothesis, it was demonstrated that total myofibrillar and sarcoplasmic protein concentrations along with total myosin and actin content were not differentially altered between or within the ‘high’ and ‘low’ responder cohorts. The authors concluded from this finding that the higher increases in muscle fiber cross-sectional with ‘high’ responders is perhaps more likely due to factors other than myofibrillar or sarcoplasmic protein concentration increases. The authors have performed a very well-designed study and investigated a unique, but justified, hypothesis. They have adequate participant numbers for their analyses and a highlight of their methods were the three measures used to generate the high and low responder cohorts (which presents a sound basis as opposed to other publications on this topic that aren’t based on such stringent measures). The paper is very well written (consistent and appropriate use of language and grammar), and my comments/ concerns are only minor in nature.

1. Regardless of the sound hypothesis stated in the manuscript, wouldn’t an increase in mitochondrial content/ function/ capacity with resistance exercise work more against the ‘specificity of adaptation’ where the muscle is more phenotypically adapted to promote molecular events promoting muscle growth (i.e.: increases in translation initiation signalling, etc) than try to promote the opposite of this?

2. Substantial detail regarding the methodology used to measure myofibrillar and sarcoplasmic protein expression is provided, however I still wonder about the sensitivity of the measure. How well optimised are these blots- for instance, have different concentrations of loaded protein been used to determine the amount of protein that provides optimal signal while still keeping sensitivity to measure differences between samples? Also, how well are antibody specificity and associated dilutions optimised?

·

Basic reporting

Well done

Experimental design

Well done overall. My one comment would be to include detailed description of the resistance training protocol. It is inconvenient for readers to have to locate another paper, and some may not have access to the paper. The RT protocol has relevance to the findings, so it should be explained.

Validity of the findings

Well done

Comments for the author

Overall, this is very well-done paper with potentially important implications to the field. The writing was excellent, the methods sound, and the conclusions appropriate. I have a few suggestions for improvement:

First, the finding of greater mitochondrial capacity is interesting, and I would agree that the higher energy cost of MPS is a potential link. However, are high responders better able to upregulate mitochondrial proteins or is this simply a consequence of a greater increase in turnover and thus a byproduct? I realize that there is no way to know, but it would be worthwhile to provide speculation here.

Second, you hypothesize that intracellular fluid accumulation may explain differences in muscle size between high and low responders. What mechanism could be responsible for such a hypothesis? It would seem counterintuitive to me that this would be explanatory, given that fluid changes are temporary and can be altered in a very short time frame (thus, the hypertrophy conceivably would not be “lasting”). If the fluid changes were somehow chronic, then I don’t see how this would occur without accompanying increases in sarcoplasmic proteins, which would necessitate greater surrounding fluid to maintain a stable milieu. I would think there could be negative consequences on the function of the cell from higher intracellular fluids without accompanying protein accretion, as this would conceivably disturb the “spacing” of tissues required for cellular balance. Again, I realize that there is no way to know the answers to these questions, but I feel they should be addressed.

---

## Round 0.2 · accepted · Accept

Congratulations on addressing the final concerns of the reviewers and myself; we are now happy to accept this manuscript for publication in PeerJ.

# ·

Basic reporting

See my general comment(s)

Experimental design

See my general comment(s)

Validity of the findings

See my general comment(s)

Comments for the author

Dear Dr Roberts and colleagues,

Thanks for the detailed responses to my comments and questions, and the other reviewers', including the provision of additional data for my own perusal.

I'm happy with the responses you've provided, and the subsequent edits made.

Llion

·

Basic reporting

No comment

Experimental design

No comment

Validity of the findings

No comment

Comments for the author

The authors have satisfactorily addressed all my comments and are to be commended on their quality of work.

·

Basic reporting

See below

Experimental design

See below

Validity of the findings

See below

Comments for the author

The authors have satisfactorily addressed my comments and I feel this will be an excellent addition to the literature.